# Postoperative Infection after Anterior Cruciate Ligament Reconstruction: Prevention and Management

**DOI:** 10.3390/microorganisms10122349

**Published:** 2022-11-28

**Authors:** George A. Komnos, George Chalatsis, Vasilios Mitrousias, Michael E. Hantes

**Affiliations:** The Department of Orthopaedic Surgery, Faculty of Medicine, School of Health Sciences, University of Hospital of Larissa, 41110 Larissa, Greece

**Keywords:** ACL, septic arthritis, infection, vancomycin, graft removal

## Abstract

Infection following anterior cruciate ligament (ACL) reconstruction can be one of the most debilitating complications following ACL reconstruction. Its reported incidence is around 1%. Utilization of vancomycin for presoaking the graft is considered an established method for infection prevention. The role of other agents, such as gentamycin needs further investigation. *Staphylococci* are the predominant causative pathogens, while particular attention should be paid to fungal infections due to their long-standing, occult process. Recent data demonstrate that hamstrings autograft may be at an elevated risk of being contaminated leading to subsequent septic arthritis. Diagnosis is set by clinical and laboratory findings and is usually confirmed by intraoperative cultures. Treatment varies, mainly depending on the intraoperative assessment. Satisfactory outcomes have been reported with both graft retaining and removal, and the decision is made upon the arthroscopic appearance of the graft and the characteristics of the infection. Of note, early management seems to lead to superior results, while persistent infection should be managed with graft removal in an attempt to protect the articular cartilage and the knee function.

## 1. Introduction

Anterior Cruciate Ligament (ACL) reconstruction remains one of the most common orthopaedic procedures, with a raising trend during the last few years. In England, its rate has been reported to be 24.1/100,000, while being 2/100/000 two decades ago [1]. This rate is even higher in Scandinavia and USA, ranging from 32 to 52/100,000, respectively [2].

The target group of patients suffering from an ACL tear and finally undergoing reconstruction is relatively young, with a mean age of 28 years [3]. Being a procedure that is predominantly applied in a young and healthy population, it is characterized by few and, on rare occasions, serious complications. Nonetheless, infection following ACL reconstruction can be one of the most debilitating complications following ACL reconstruction. Its incidence, besides being low, is not negligence, with the reported incidence rates being less than 1% [4,5,6,7,8]. Slightly higher rates have been also reported, ranging from 1.4 to 2% [9,10,11].

Epidemiologic data of pediatric and adolescent patients have shown that the infection rate in this population is comparable to adults. In particular, Eisenberg et al. analyzed a large sample of 44,501 patients below 18 years old and revealed an infection rate of 0.52% [12]. This was comparable to the infection rate of the control group (adults, aged 18–30, 0.46%). Remarkably, patients below 15 years old had a significantly lower rate of infection compared with adolescents (15 to 17 years old).

Despite rare, infection following ACL reconstruction can lead to an acute or gradual knee dysfunction, provoking cartilage damage and graft failure [13,14]. Moreover, patients who go through this condition are most likely to undergo multiple procedures and have poorer final outcomes [15]. 

The aim of this review is to present an up-to-date overview of graft differences, causative pathogens, preventive measures, and therapeutic approaches in ACL reconstructions complicated with infection in an attempt to achieve lower infection rates and successfully manage this severe complication.

## 2. Graft Differences

In general, allografts have been initially linked with a higher infection rate in comparison with autografts [16]. This was the case in the early use of allografts, which were accused of viral and bacterial transmission. However, in the last years, novel techniques of irradiation and chemical processing have overcome this situation [17].

In the ACL surgery setting, published data are not in absolute agreement, with conflicting reports of equal infection rates between autografts and allografts, or higher rates of allografts [5,6,18]. Nonetheless, recent data support the equality of autografts and allografts in terms of provoking postoperative infection. Greenberg et al. [17] found no increased clinical risk of infection with the use of allograft tissue compared with autologous tissue for primary anterior cruciate ligament reconstruction. Katz et al. ended up with the same conclusion, failing to identify a higher rate of deep bacterial infection when allograft tissue was used [19]. 

Regarding autografts, recent data have raised concerns regarding the utilization of hamstrings tendon autografts, identifying it as a risk factor for postoperative septic arthritis. Several studies have found hamstrings autograft to be more susceptible to postoperative infection than bone-patella bone-tendon (BPTB) autograft [6,7,18,20]. Marom et al. [8] in one of the largest series dealing with infection following ACL procedures, demonstrated an increased incidence of infections in patients with hamstrings autograft, or allograft compared to BPTB autograft. Above that, hamstrings autograft was linked to a higher possibility of infection in contrast to all other grafts, including the patellar tendon, the quadriceps tendon, and allografts [5]. In another retrospective study of 3126 procedures, hamstrings autograft revealed an elevated incidence of infection than both BPTB autograft and allograft, with a trend towards a higher need for graft removal [18].

In a large nationwide research study, hamstrings autograft was found to be an independent risk factor of septic arthritis following ACL reconstruction [3]. Finally, in a Meta-Analysis of 198 Studies with 68,453 grafts, it was found that hamstrings autografts were associated with a higher risk for infection that BPTB autografts and allografts [21]. More specifically, hamstrings autografts presented with an infection rate of 1.1%, BPTB autografts 0.7%, and allografts 0.5%.

Parameters that make hamstrings vulnerable to infection have not been well studied yet, with contamination after harvesting and during the preparation of the graft being the most accepted hypothesis [7,22].

## 3. Microbiology

The most common pathogens causing septic arthritis following ACL reconstruction are *Staphylococcus aureus* (*S. aureus)* and *Coagulase-negative Staphylococcal* species. The last ones mainly include *Staphylococcus epidermidis* followed by *Cutibacterium acnes* (former *Propionibacterium acnes)* [15,23,24,25]. In one of the largest relevant cohorts, Marom et al. [8] identified in 11,451 procedures the *coagulase-negative staphylococci* as the most common pathogen, followed by *methicillin-susceptible Staphylococcus aureus* (MSSA). In our institutional setting, *Coagulase-negative Staphylococcal and S. aureus* were the isolated pathogens in all infected cases (5 and 2, respectively) in a total of about 2000 procedures [20].

*Enterobacter* spp. have also been isolated, which are treatable with gentamicin [4], while fungal infections are rare, with few reports existing so far in the literature [26,27]. Interestingly, in one of the reported cases with a fungus infection presoaking of the graft had been performed [26].

As aforementioned, fungal infections have been also described but in a very limited number. Among others, *Rhizopus microspores*, *Candida albicans*, and *Aspergillus species* have been demonstrated [26,28,29]. These infections are characterized by an indolent course that makes them extremely difficult to diagnose. The most worrying issue with fungal infections is that their indolent course provokes chondrolysis and remarkable bone loss that has to be appropriately addressed during the revision procedure. A high index of suspicion and early removal of all implanted materials and grafts are of paramount importance in such cases. Noteworthy, fungal infections may need advanced techniques for diagnosing, such as PCR and MALDI-TOF, since conventional cultures miss to diagnose them [26]. Costa-Paz et al. [30] published a case series of 21 patients with a fungal (mucormycotic) infection after ACL reconstruction [30]. They highlighted the severity of those cases, which were characterized by occult infections that finally resulted in exceptional bone loss. Astonishingly, the majority of the patients developed neglected septic arthritis and osteomyelitis that ultimately destroyed the knee joint, while one of them had to undergo an above-knee amputation, due to a severe septic shock. 

Scarce pathogens have been also isolated in cases with septic arthritis in ACL settings. One of those is *Clostridioides difficile* which contaminated an immunocompetent patient who underwent ACL reconstruction with a hamstrings autograft [31]. 

## 4. Prevention

One of the most popular issues regarding infection following ACL reconstruction, in recent literature, is the use of antibiotic agents, especially vancomycin for presoaking the grafts, as a successful measure for infection prevention. Apart from intraoperative intravenous (IV) antibiotics, which remain the gold standard for preventing infections during surgical procedures, presoaking the graft with an antibiotic agent seems to be an efficient strategy to prevent postoperative deep infections. The most popular and effective agent until now, is vancomycin. Several studies have demonstrated a decrease or even elimination after the application of a protocol of presoaking the graft with vancomycin [32,33,34]. Our group’s published experience highlights the effectiveness of the utilization of vancomycin [33]. Pre-soaking the graft with vancomycin reduced the rate of postoperative infections in ACL reconstructions from 0,56% (7/1242 cases) to 0% (0/593 cases).

The proposed concentration of vancomycin is 5 mg/mL [11]. Nevertheless, the presoaking time period has not been standardized yet. A minimum of 15 min has been suggested by Figeroa et al. [35] while Schüttler et al. in a laboratory study proposed a minimum presoaking of 20 min [36]. Xiao et al. in a systematic review found that presoaking the ACL tendon grafts with vancomycin leads to a nearly 15 times decrease in the odds of infection [37]. Among others, vancomycin has been proven to be highly cost-effective compared with isolated prophylactic antibiotic administration [38]. Nonetheless, patient-reported outcomes and retear rates after soaking ACL grafts in vancomycin have not been studied in-depth. 

Another agent that has been proposed for the prevention of infection is gentamycin. Moriarty et al. applied a gentamicin concentration of 0.8 mg/mL in hamstrings autografts and found a decrease in cases of postoperative septic arthritis but no difference in superficial surgical infections [4]. However, there is a paucity of data on whether there are differences in efficacy, side effects, and superiority between gentamycin and vancomycin.

Ultimately, proper skin preparation remains of paramount importance, since grafts have been proven to be principally contaminated by the patient’s skin commensal organisms (24).

## 5. Management-Therapeutic Approach

### 5.1. Diagnosis

Diagnosis in patients who are suffering from infection after an ACL procedure is relatively straightforward. Clinical features consist of fever, fatigue, pain, and swelling of the knee, which is notably usually warm. In the vast majority of cases, symptoms onset is between 10 to 25 days after ACL reconstruction. Laboratory findings include elevated C-reactive protein (CRP) and erythrocyte sedimentation rate (ESR). A patient who presents with the above characteristics undergoes knee aspiration. The aspirated fluid is sent for analysis and cultivation. Costa et al. identified synovial white blood cell (WBC) count as the most reliable marker for the diagnosis of septic arthritis after ACL reconstruction [39]. A cut-off value of 28.000 cells/mL has been recently proposed to provide the highest accuracy [39].

### 5.2. Initial Management

After setting the diagnosis of the infection, most protocols include early antibiotic administration combined with surgical management with arthroscopic irrigation and debridement (I&D), which has shown favorable outcomes [40,41]. Initially, empirical intravenous antibiotic therapy is administrated. This usually consists of a combination between ceftazidime (2 g/8 h), vancomycin (1 g/12 h), cefazolin, flucloxacillin (6 × 1 g/d), and gentamycin (320 mg/d) [24,42,43]. Early intervention plays an essential role to achieve the desired outcome. Clinical suspicion of infection should lead to an arthroscopic lavage and inspection of the graft.

### 5.3. Therapeutic-Surgical Issues

Different management methods exist, including intravenous (IV) antibiotics, multiple reoperations for irrigation and debridement, graft removal, and staged revision reconstruction [24]. There are two main concerns besides the eradication of the infection. These consist of the number of arthroscopic debridements that are necessary and the retention or not of the graft. During the arthroscopy, the surgeon evaluates the condition of the graft and decides whether this looks healthy, so it can be retained, or not. Cultures of the synovial tissue have to be collected and sent for cultivation to identify the causing pathogen. Graft retention rates range from 63% to 100% [8,15,44,45]. Unfortunately, there are no established guidelines for retaining or not the graft, and this relies on the surgeon’s judgment, taking into consideration the arthroscopic visualization of the graft, along with the clinical and laboratory findings. 

Categorizing the therapeutic approaches, we could divide them into 3 categories: (1) graft retention with repetitive, if necessary, arthroscopic debridements, (2) graft removal after the failure of initial arthroscopic debridement, and (3) immediate graft removal.

Several published studies are in favor of graft retention, applying arthroscopic lavage and debridement accompanied by antibiotic suppression. Success rates of as much as 85% have been reported with this process [46]. Utilization of autograft has been proposed to be in favor of graft retaining [47]. When initial lavage fails to eradicate the infection, then removal of the graft should be strongly considered. Three or more I&D procedures have been correlated with ultimate graft removal [8]. Similar recommendations were provided by a consensus statement of Indian orthopaedic arthroscopy surgeons [48]. They recommended the removal of the graft when repeated arthroscopic debridements are required. Hantes et al. [20] recommended a low threshold in debridements before deciding to remove the graft. Protection of the articular cartilage and maintenance of good knee function is of higher importance than retainment of the graft, which can be reconstructed later on with a revision procedure. Failure of the initial debridements probably implies the formation of biofilm that predisposes to the persistence of the infection. We have also to bear in mind that patients who sustain a considerable amount of reoperations appear to suffer from ligamentous laxity, even after ACL revision [45]. Acute removal is reserved for cases of **bone** involvement, damaged graft, and severe infection [49,50,51]. Kusnezov et al. [45] in a systematic review advocated for a low threshold in graft removal even in the index procedure, especially when intraoperative concerns about the appearance of the graft arise.

As aforementioned, graft retention, when feasible, is essential for achieving better functional results [45]. Nevertheless, a recent study showed no statistically significant differences in the Lysholm or the IKDC score, between patients who had their graft initially retained or initially removed and then underwent a revision procedure [41]. Above that, the presence of graft at the final follow-up after successful treatment of infection resulted in statistically superior outcomes on the IKDC score but not on the Lysholm score.

### 5.4. Post-Operative Antibiotics

Another issue of debate is the duration of antibiotic administration. Initial intravenous therapy is obligatory and lasts 2 to 4 weeks [8,41]. Afterward, intravenous antibiotics change to culture-sensitive oral antibiotics for at least 3 weeks, or until normalization of clinical and laboratory parameters [20,42]. The antibiotic medication is based on the sensitivity of the microorganism cultivated from the aspiration, or the intraoperative cultures. Suppression for a 12-week period has been also proposed so as to achieve further incorporation and vascularization of the graft [8]. A prolonged duration of antibiotics is suggested in immunocompromised patients [48].

### 5.5. Initial Management Failure–Persistent Infection

Rarely, when arthroscopic management fails, open revision can be applied. Prolonged infection can occasionally have prejudicious results, and arthroscopic debridement and irrigation may not be enough. In such cases with a persistent infection that can provoke osteomyelitis in the tibial side, with concurrent soft tissue issues, debridement with a medial gastrocnemius flap, performed either by a specialized orthopaedic surgeon, or a plastic surgeon, has been suggested as a viable option to eradicate the infection [52].

To sum up, it is worth noting that in the majority of cases, this complication does not lead to catastrophic outcomes; the treatment may be extremely challenging though.

## 6. Discussion

Given the rarity of infection following ACL procedures make the understanding and management of this condition quite strenuous. The scarce nature of this complication raises difficulties in detecting risk factors that could be encountered in an attempt to eliminate postoperative infections. Nonetheless, limited data have incriminated Diabetes Mellitus as a risk factor for infection after ACL reconstruction [6]. The best-applied method in the prevention of this complication, is, until now, the utilization of antibiotic agents for soaking the grafts, before implanting them. It is nowadays well-documented that vancomycin presoaking of ACL grafts is a biomechanically sound, clinically efficient, and cost-effective method of preventing ACL reconstruction postoperative infection (52). On the contrary, concerns have been raised over cost, allergies, and the development of resistance. These parameters along with the biomechanical effect of the antibiotic agent on the long-term function of the graft, possible negative effects on articular cartilage, and a possible alteration of bacterial profile in surgical site infections require further evaluation [53,54]. Identifying risk factors for developing a postoperative infection and evaluating the effectiveness of other agents besides vancomycin should be targets of future research for prevention.

The main source of infection following ACL reconstructions have not been identified yet, but direct graft contamination seems to be the most likely causative factor [24]. This highlights the importance of keeping contact with the graft with the skin to a minimum. Another significant issue is the preparation of the graft. A high rate of contamination of 12% has been found during graft preparation [55]. Among grafts, hamstrings tendon autograft has been associated with higher infection rates. That fact has significant implications, given that it remains the most commonly harvested graft type. 

Based on the above, prevention of infection in ACL reconstruction could be summarized in the following recommendations: Administration of prophylactic intravenous antibiotics preoperatively; meticulous skin preparation and disinfection; quick and careful preparation of the graft; utilization of patellar tendon or quadriceps autograft, as their use seems to decrease the possibility of postoperative infection and vancomycin presoaking of grafts, especially when hamstrings autografts are utilized [6,33,48].

Treatment of an infected knee following an ACL procedure can be extremely challenging and probably the most debatable issue. The main goal is the complete eradication of the infection and protection of the articular cartilage through arthroscopic debridement and antibiotic suppression. A secondary but equally important goal is the accomplishment of a fully functional knee after treating the infection. The persistence of infection could have detrimental effects on the cartilage and leads to arthrofibrosis and subsequent stiffness [56]. Successful treatment of these cases results in the restoration of the patient’s activities with a safe return to work and sports activities [41]. Nevertheless, the patients should be properly counseled to lower their expectations, as return rates at the pre-injury level and frequency are relatively low. Of note, a collaboration between the orthopaedic surgeon and infectious disease (ID) specialists is of paramount importance to achieve the best outcome. Following the surgical management of the infection, ID specialists can contribute, based on the antibiograms from the cultures, to the decision for the specific antibiotic agent and the duration of the administration.

Our institutional published protocol for these cases is in accordance with published data and consists of arthroscopic debridement and irrigation of the knee joint immediately after diagnosing the infection. In case of recurrence, subsequent knee irrigation is performed with hardware and graft removal while re-implantation is proposed for patients at a later stage [20]. This protocol has shown satisfactory results in terms of knee function, and range of motion. IKDC and Lysholm scores, without rebounding in postoperative arthritis. 

Retention or removal of the graft relies on arthroscopic findings, on whether the graft looks healthy or not. Unfortunately, this parameter is extremely subjective, and relies on the surgeon’s evaluation and experience. There are no specific data or guidelines to define a graft as healthy or not. Usually, its continuity and strength are evaluated and determine whether this is considered an intact-healthy graft or not. Marom et al. attempted to identify risk factors against graft retention [8]. They revealed that older patients with higher BMI and larger synovial white blood cell count in initial joint aspiration are at an elevated risk for graft removal, but without achieving statistical significance. Notably, they observed that the increasing number of I&D procedures was associated with graft removal. Graft reimplantation can be offered to the patient when the infection retreats, at a time period ranging from 1 to 9 months, with satisfactory reported outcomes [23,45,57]

In conclusion, infection following an ACL procedure can be an extremely troublesome situation for both the patient and the surgeon. Early diagnosis and intervention are of paramount importance. The surgeon has to be aware of all available preventative measures and of the most usual risk factors and causative pathogens that can provoke those infections so as to be able to successfully decrease the infection rates. When signs of infection appear, arthroscopic debridement should be immediately performed to remove the infected tissues and visualize and evaluate the condition of the graft. Retention or removal of the graft is still an issue and relies on the surgeon’s preference, experience, and judgment. Nevertheless, the surgeon should not be afraid of removing the graft, if this appears infected, or if arthroscopic lavage and debridement do not result in the eradication of the infection. In those cases, biofilm formation is probably the reason for the recurrence of the infection, and the removal of the graft and all materials can achieve successful outcomes, in terms of infection control and functional outcomes.

## 7. Conclusions

Despite being an extremely severe and potentially disastrous complication, infection after ACL reconstruction can be properly managed by applying proper, individualized protocols. Future perspectives should focus on identifying the exact risk factors, and the most reliable preventive measures in an attempt to lower the infection rates. Furthermore, to establish the most optimal surgical management that can successfully face the infection when this appears.

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
