# Peer review of "Postoperative Infection after Anterior Cruciate Ligament Reconstruction: Prevention and Management"

_microorganisms, 2022, doi:10.3390/microorganisms10122349_

Round 1
Reviewer 1 Report
The subject dealt with is of clinical relevance especially in the orthopedic field. Infections are frequent and their management is not easy. The review is well written but totally missing the part on the preoperative, intraoperative and post operative management of the surgical site. In fact, there are strategies that do not use systemic antibiotics with local application and that can be implemented and here. I do not see any mention of them. I therefore consider it necessary to add these parts.Author Response
Reviewer 1
The subject dealt with is of clinical relevance especially in the orthopedic field. Infections are frequent and their management is not easy. The review is well written but totally missing the part on the preoperative, intraoperative and post operative management of the surgical site. In fact, there are strategies that do not use systemic antibiotics with local application and that can be implemented and here. I do not see any mention of them. I therefore consider it necessary to add these parts.
Response
Thank you for your time spent to improve our manuscript and your suggestions. Regarding the management of the surgical site on the preoperative part, we believe that this is covered with the part of prevention, the intraoperative with the main surgical issue that exists, the retention of the graft, and postoperative, with the time of the antibiotics administration. If you would like to specify more the term surgical site and you believe that we miss something more we would be more than willing to add it. Nevertheless, upon your request and other reviewer’s request more data have been added about antibiotic treatment.
Reviewer 2 Report
The manuscript “Postoperative infection after anterior cruciate ligament reconstruction: prevention and management” is a narrative review on this post-surgical complication, with details on epidemiology, characteristics, microbiology, prevention and treatment. The manuscript is of interest for clinicians and readers interested in bone and joint infections, from orthopaedic surgeons to clinical microbiologists and infectious diseases specialists. In general terms, it is well written, despite the fact that some English expressions may be considered for revision. There are some major points that, if addressed, could make the article suitable for a new revision.
Firstly, the discussion section is repetitive and, basically, includes more or less the same information that has been mentioned previously in the different sections of the manuscript. It adds very little to what has been included. Having said that, I strongly encourage the authors to reconsider this section and modify it to a discussion on areas of uncertainty or future research directions, if considered appropriate.
Secondly, the section on management-therapeutic approach is disorganized and difficult to follow. There are paragraphs on diagnosis, surgical treatment options, antimicrobial therapy… I believe this section may benefit from subheadings or subsections. Similarly, as mentioned above, I would suggest that repetitive information might be avoided or included together in similar paragraphs or subsections. In contrast, readers might appreciate it if more data on the antimicrobial treatment was included, such as empirical treatments or targeted therapies (a table might help?). In addition, there is no mention of whether anti-biofilm antibiotics, such as rifampin, are needed or not in this setting.
Finally, the section on Microbiology seems a bit imbalanced. A great deal of attention is paid to the fungal aetiology, although it is a very rare isolation, whereas only two short paragraphs are focused on the main microorganisms causing these infections. A reader who is not familiar with these infectious disease may have the idea that fungal infections are more frequent that what they actually are. Other data may include methods of microbiological sampling, culture methods or discuss new methods of microbiological procedures in general (such as PCR and others).
There are several minor points that should also be addressed:
- - Please follow international recommendations on naming microorganisms. Examples: Line 86 Staphylococcus aureus (S. aureus subsequently in the text, if needed); Line 87 Cutibacterium acnes (in italics, acnes not with a in capital letters). Etc…
- - Antibiotics such as vancomycin can be written in the middle of the text without a capital letter (instead of Vancomycin).
- - Line 8: Define ACL before abbreviation.
- - Line 9: Its reported incidence is around 1%.
- - Line 27: rates instead of incidence?
- - Lines 33-34: “following ACL reconstruction” is repeated in the sentence.
- - When showing numbers in the text, please do not use commas for decimals (4.6 instead of 4,6).
- - Line 67: What are BPTB? Define before abbreviating.
- - Line 90: Are authors here referring to Enterobacteriaceae, Gram negative microorganisms or Enterobacter spp.? Please clarify and name appropriately.
- - Line 109: I believe that the authors refer to rare pathogens instead of scarce pathogens. Is that so?
- - Lines 119-120: Decrease or elimination of incidence? These sentence may be rephrased. I suggest another word rather than elimination should be used.
- - Regarding the use of vancomycin to presoak the grafts, has this been associated with infections by Gram-negative microorganisms? What was the microbiology of infections in those patients in whom vancomycin presoaking was employed?
- - Line 147: “A cut-off value of 28.000 cells/ml has been proved to provide the highest accuracy[42]”. Only one article is mentioned. Please provide more data or reduce the strenght of the sentence.
- - Lines 150-151: “Nevertheless, septic arthritis following ACL reconstruction remains a relatively rare complication that is difficult to manage.” I do not understand this sentence. As it is written, the authors suggest that infections following ACL infection may associate with septic arthritis or not. This is not in line with what has been mentioned before in terms of diagnosis (knee aspiration) or treatment (arthroscopic debridement). I find it difficult not to label the post-surgical infection following ACL reconstruction as a septic arthritis. Please justify.
- - Lines 155-156: “The main concerns besides the eradication of the infection are the number of arthroscopic debridements that have to be done and especially whether the graft has to be retained or not”. Please rephrase.
- - Line 160: What are the criteria to consider that a graft looks healthy? It is just dependent on the surgeon or are there specific criteria?
- - Line 177: How low should the threshold be before removing the graft? Please provide more specific data.
- - Line 184: bone or bony?
- - Line 185: Again, how low should the threshold be?
- - What is the opinion of the authors regarding the duration of antimicrobial treatment? Some authors usually treat these infections between 3-6 weeks instead of prolonged durations, as mentioned in the manuscript? Are suppressive antibiotic therapies really needed?
- - The importance of a infectious disease specialist in a multidisciplinary management of these infections may be worth mentioning, too.
- - Line 201: “…debridement with a medial gastrocnemius flap…”. Probably needed if there are problems with the soft tissues and plastic surgeons may also collaborate. Please rephrase.
Author Response
Reviewer 2
The manuscript “Postoperative infection after anterior cruciate ligament reconstruction: prevention and management” is a narrative review on this post-surgical complication, with details on epidemiology, characteristics, microbiology, prevention and treatment. The manuscript is of interest for clinicians and readers interested in bone and joint infections, from orthopaedic surgeons to clinical microbiologists and infectious diseases specialists. In general terms, it is well written, despite the fact that some English expressions may be considered for revision. There are some major points that, if addressed, could make the article suitable for a new revision.
- Firstly, the discussion section is repetitive and, basically, includes more or less the same information that has been mentioned previously in the different sections of the manuscript. It adds very little to what has been included. Having said that, I strongly encourage the authors to reconsider this section and modify it to a discussion on areas of uncertainty or future research directions, if considered appropriate.
Initially, we would like to thank you for the effort you put to review and meliorate our manuscript. Discussion is now modified according to your suggestion.
- Secondly, the section on management-therapeutic approach is disorganized and difficult to follow. There are paragraphs on diagnosis, surgical treatment options, antimicrobial therapy… I believe this section may benefit from subheadings or subsections. Similarly, as mentioned above, I would suggest that repetitive information might be avoided or included together in similar paragraphs or subsections. In contrast, readers might appreciate it if more data on the antimicrobial treatment was included, such as empirical treatments or targeted therapies (a table might help?). In addition, there is no mention of whether anti-biofilm antibiotics, such as rifampin, are needed or not in this setting.
Thank you for raising these important issues. Subheadings have been added. Some sentences have been deleted or rearranged to avoid repetitive information. Data regarding antibiotic treatment have been added. Finally, regarding rifampin, as biofilm and infection following ACL reconstructions are established commonly in the autograft and not in materials, rifampin seems that has no significant role in this area.
- Finally, the section on Microbiology seems a bit imbalanced. A great deal of attention is paid to the fungal aetiology, although it is a very rare isolation, whereas only two short paragraphs are focused on the main microorganisms causing these infections. A reader who is not familiar with these infectious disease may have the idea that fungal infections are more frequent that what they actually are. Other data may include methods of microbiological sampling, culture methods or discuss new methods of microbiological procedures in general (such as PCR and others).
Thank you for your comments. Maybe wrong, but we took for granted that more readers would be aware of the most common microorganisms and tried to raise awareness of these rare situations. We tried to make clear that these are dangerous but very infrequent infections.
- There are several minor points that should also be addressed:
- - Please follow international recommendations on naming microorganisms. Examples: Line 86 Staphylococcus aureus (S. aureus subsequently in the text, if needed); Line 87 Cutibacterium acnes (in italics, acnes not with a in capital letters). Etc…
Thank you for your comment. These have now changed.
- - Antibiotics such as vancomycin can be written in the middle of the text without a capital letter (instead of Vancomycin).
Thank you for highlighting this issue. Vanvomycin in now with lower v throughout the manuscript
- - Line 8: Define ACL before abbreviation.
This is now defined
- - Line 9: Its reported incidence is around 1%.
This is corrected
- - Line 27: rates instead of incidence?
Thank you. That is correct. The term rate has been added instead of incidence
- - Lines 33-34: “following ACL reconstruction” is repeated in the sentence.
Thank you. The one sentence has been erased
- - When showing numbers in the text, please do not use commas for decimals (4.6 instead of 4,6).
This is now corrected
- - Line 67: What are BPTB? Define before abbreviating.
Thank you. This is now defined
- - Line 90: Are authors here referring to Enterobacteriaceae, Gram negative microorganisms or Enterobacter spp.? Please clarify and name appropriately.
This is now mentioned as Enterobacter spp
- - Line 109: I believe that the authors refer to rare pathogens instead of scarce pathogens. Is that so?
We believe that the meaning is the same. But if the reviewer insists we can change the word scarce with rare.
- - Lines 119-120: Decrease or elimination of incidence? These sentence may be rephrased. I suggest another word rather than elimination should be used.
Thank you for raising this issue. Published data, and our published experience has shown that infections rates have been eliminated (have reached 0%) after vancomycin application. Therefore, we intentionally used this phrase to show that infections following ACL reconstructions can disappear with utilisation of vancomycin
- - Regarding the use of vancomycin to presoak the grafts, has this been associated with infections by Gram-negative microorganisms? What was the microbiology of infections in those patients in whom vancomycin presoaking was employed?
Presoaking the graft with vancomycin is a means of prevention, and not treatment. Therefore, we do not know the microbiology of the infections at that time, neither if this particular knee will be infected. Vancomycin, on the other hand, has been proposed because the most common species that are responsible for this kind of infections are gram positive ones.
- - Line 147: “A cut-off value of 28.000 cells/ml has been proved to provide the highest accuracy[42]”. Only one article is mentioned. Please provide more data or reduce the strength of the sentence.
This sentence has been rephrased accordingly. “recently proposed” instead of “proven”
- - Lines 150-151: “Nevertheless, septic arthritis following ACL reconstruction remains a relatively rare complication that is difficult to manage.” I do not understand this sentence. As it is written, the authors suggest that infections following ACL infection may associate with septic arthritis or not. This is not in line with what has been mentioned before in terms of diagnosis (knee aspiration) or treatment (arthroscopic debridement). I find it difficult not to label the post-surgical infection following ACL reconstruction as a septic arthritis. Please justify.
We apologise if something is not clear. But septic arthritis and infection following ACL reconstruction have actually the same meaning
- - Lines 155-156: “The main concerns besides the eradication of the infection are the number of arthroscopic debridements that have to be done and especially whether the graft has to be retained or not”. Please rephrase.
Thank you for raising this point. This is now rephrased.
- - Line 160: What are the criteria to consider that a graft looks healthy? It is just dependent on the surgeon or are there specific criteria?
Thank you for raising this issue. This is now made more clear through a comment - clarification that is added in the discussion part (lines 252-255)
- - Line 177: How low should the threshold be before removing the graft? Please provide more specific data.
Thank you for this comment. As written in the manuscript, the number of arthroscopic procedures for lavage is a question of debate in orthopaedic society with no established number-recommendation. There is no defined threshold to add, but as we mention published data and our experience are in favor of low threshold (one lavage procedure).
- - Line 184: bone or bony?
Thank you. Bony is now altered with bone
- - Line 185: Again, how low should the threshold be?
This is answered in the previous question
- - What is the opinion of the authors regarding the duration of antimicrobial treatment? Some authors usually treat these infections between 3-6 weeks instead of prolonged durations, as mentioned in the manuscript? Are suppressive antibiotic therapies really needed?
As published from our department, we usually propose a 3-week administration or until normalization of the inflammatory markers. This is now made clear in the part of post-operarative antibiotics. Suppressive antibiotic therapies are reserved for specific, exceptional cases.
- - The importance of a infectious disease specialist in a multidisciplinary management of these infections may be worth mentioning, too.
Thank you for mentioning that and we apologise for omitting this information. This is now added in the discussion. (Lines 243-247)
- - Line 201: “…debridement with a medial gastrocnemius flap…”. Probably needed if there are problems with the soft tissues and plastic surgeons may also collaborate. Please rephrase.
This is now rephrased upon your comments
Reviewer 3 Report
Authors have described the main tools for prevention and management of postoperative infection after anterior cruciate ligament reconstruction. They include a large list of references and address the main issues of ACL postoperative infections.
Although the topic is of interest for the scientific community, I would have liked to have seen in the manuscript additional concepts regarding personal experiences as well as a critical evaluation of the literature, and not a simple description of the various issues (already confimed by the literature). I would like to encourage the authors to rewrite the article in this way, trying to give a very practical knowledge pills for the surgeon, the ID specialist or Microbiologist.
Author Response
Reviewer 3
Authors have described the main tools for prevention and management of postoperative infection after anterior cruciate ligament reconstruction. They include a large list of references and address the main issues of ACL postoperative infections.
Although the topic is of interest for the scientific community, I would have liked to have seen in the manuscript additional concepts regarding personal experiences as well as a critical evaluation of the literature, and not a simple description of the various issues (already confimed by the literature). I would like to encourage the authors to rewrite the article in this way, trying to give a very practical knowledge pills for the surgeon, the ID specialist or Microbiologist.
Response
Thank you for your effort and your remarks on the manuscript. Several sentences have been added in the manuscript to highlight these issues and present the issue in the way you and the other reviewers propose.
Reviewer 4 Report
Introduction needs to be improved. Such as, the describe and presentation in line 27 and 28 is hard to read and understand. Dot and comma are mixed. Same problems can be found in line 36, line 39, and line 147. Line 85 to line 89 need to be revised.
Author Response
Reviewer 4
Introduction needs to be improved. Such as, the describe and presentation in line 27 and 28 is hard to read and understand. Dot and comma are mixed. Same problems can be found in line 36, line 39, and line 147. Line 85 to line 89 need to be revised.
Thank you for your effort and your comments
Some commas have been erased, and other modifications have been also made in this parts upon your comments.
Round 2
Reviewer 2 Report
I acknowledge the effort of the authors to meet this reviewer's recommendations and suggestions. The manuscript is clearly improved with these changes.
Reviewer 4 Report
Improved manuscript.